# Safety and Feasibility Study of the Medical Care Pit Walking Support System for Rehabilitation of Acute Stroke Patients

**DOI:** 10.3390/jcm12165389

**Published:** 2023-08-19

**Authors:** Hiroki Watanabe, Bryan J. Mathis, Tomoyuki Ueno, Masakazu Taketomi, Shigeki Kubota, Aiki Marushima, Hiroaki Kawamoto, Yoshiyuki Sankai, Akira Matsumura, Yasushi Hada

**Affiliations:** 1Department of Neurosurgery, Institute of Medicine, University of Tsukuba, Tsukuba 305-8575, Ibaraki, Japan; watanabe.hiroki.gb@u.tsukuba.ac.jp (H.W.);; 2International Medical Center, University of Tsukuba Hospital, Tsukuba 305-8576, Ibaraki, Japan; 3Department of Rehabilitation Medicine, Institute of Medicine, University of Tsukuba, Tsukuba 305-8575, Ibaraki, Japan; 4Department of Occupational Therapy, Ibaraki Prefectural University of Health Sciences, Ami 300-0394, Ibaraki, Japan; 5Institute of Systems and Information Engineering, University of Tsukuba, Tsukuba 305-8573, Ibaraki, Japan; 6Ibaraki Prefectural University of Health Sciences, Ami 300-0394, Ibaraki, Japan

**Keywords:** Medical Care Pit, walking support system, walking, physical therapy, acute stroke, stroke rehabilitation, Functional Ambulation Category (FAC), independent walking, gait parameters, care burden of therapy staff

## Abstract

Stroke rehabilitation with mechanical assistance improves outcomes by facilitating repetition and relieving the care burden of therapy staff. Here, we tested the Medical Care Pit (MCP) walking assistance training device in the rehabilitation of eight acute stroke patients (median age 60.7 ± 16.3 years) who had recently suffered ischemic (three) or hemorrhagic (five) stroke (14.1 ± 6.5 days). Patients received standard rehabilitation approximately 5 days per week (weekdays only), plus MCP therapy twice a week, totaling four MCP sessions over 2 weeks. Fugl–Meyer Assessment-Lower Extremities (FMA-LE), Functional Ambulation Category (FAC), and other gait-associated parameters were measured. Over the 10.5 ± 1.6 days of therapy, MCP qualitatively assisted in gait analysis and real-time patient feedback while independent walking scores significantly improved (FAC 2.2 ± 0.8 to 3.1 ± 1.3, *p* = 0.020). FMA-LE scores also slightly improved but not to significance (*p* = 0.106). Objective burden on patients, as measured by modified Borg scale, was significantly improved (2.7 ± 1.6 to 2.0 ± 1.6, *p* = 0.014). In terms of questionnaires, anxiety scores for the physical therapist regarding gait training and falling with MCP significantly decreased (3.8 ± 2.3 to 1.0 ± 1.6; *p* = 0.027 and 3.1 ± 2.2 to 0.8 ± 1.3; *p* = 0.045) from the first to fourth sessions. Taken together, MCP, in addition to the usual rehabilitation program, was effective in gait rehabilitation for independent walking and relieved burdens on the patients. Such walking support systems may be an important part of acute stroke rehabilitation.

## 1. Introduction

With a global, age-adjusted incidence of 150.77 per 100,000 people per year, stroke burdens survivors with a grueling and often frustratingly slow recovery process while neuromuscular training and rehabilitation attempts to rewire the brain by forming new neurons to recover somatic functionality [1]. Stroke rehabilitation is crucial to reduce the effects of after-stroke onset as well as increase activities of daily living (ADL) and independent walking [2]. Rehabilitation for stroke survivors entails exercises and activities designed to maximize the effect of neuroplasticity. Very early intervention (mobilization, active exercise, and intensive gait training) is therefore essential to improve body function, ADL, and walking ability in acute stroke patients [3,4]. Since 5-year outcomes for ischemic stroke are dismal (70.6% death or full dependency), especially in the elderly, electromyography, virtual reality (VR), and robotic-assisted rehabilitative techniques to maximize arm and leg recovery potential have been extensively tested and reported [5,6,7,8,9]. Of importance, the development of adaptive exoskeletons for the hand to more rapidly develop motor function has paved the way for larger and more comprehensive walking-centered systems [10].

The Medical Care Pit (MCP) system was designed by Cyberdyne, Inc. to assist and analyze gait and walking movements (Figure 1). The benefits of introducing this device into stroke rehabilitation, gait assessment, and practice include (1) constant unloading (reduction in weight bearing) and independent left/right gait control using real-time monitoring and automatic unloading adjustment, (2) automatic recording of gait measurements to increase monitoring efficiency, (3) visual feedback of gait analysis (i.e., step-length symmetry and toe clearance during walking), and (4) a harness (trunk belt) to enable the repetition of active walking movements while preventing falls.

## 2. Materials and Methods

### 2.1. Study Design

This was an interventional, single-center, single-arm, open-label, no-treatment control study.

### 2.2. Participants

Eight patients with acute stroke participated in this study. This study was conducted at the University of Tsukuba Hospital between October 2018 and December 2019. All patients had acute stroke due to cerebral infarction or cerebral hemorrhage and had adversely affected standing and/or walking ability. Patients who satisfied all the following criteria were targeted: (1) ability to give written consent in person (a substitute was appointed if the patient had difficulty in writing due to hemiplegia, etc.), (2) age of 20 years or older at acquisition of informed consent, (3) six months or less after stroke onset, (4) reduced walking and standing capacity (scores of 0 to 4 on the Functional Ambulation Category), (5) a height between approximately 140 to 190 cm plus a body weight between approximately 40 to 120 kg, and (6) fully available for hospitalization during the study period.

Exclusion criteria were as follows: (1) presence of high-grade akinesia, rigidity, and/or ataxia, (2) inability to visit the rehabilitation room, (3) an inability to understand commands or communicate, (4) the presence of high-grade muscle spasms, deformity, and/or contracture on the lower extremity, (5) implanted and active medical devices (such as cardiac pacemakers), (6) pregnancy, and/or (7) judged by doctors to be medically unstable after a comprehensive review of physical findings, blood test results, etc.

### 2.3. Study Schedule

There were 3 periods in the study schedule. The first was the pre-observation period, during which the rehabilitation doctor and physical therapist confirmed that the patients were eligible for all criteria. After the eligibility process and consent to participate were finished, the patient completed the pre-assessments. The patients conducted traditional physical therapy, with occupational and/or speech-language-hearing therapies also performed as necessary.

The second phase was traditional therapy, consisting of range of motion and muscle strength exercises, endurance work, and coordination of movement in addition to active movement facilitation practice, stretching of the trunk and four extremities, basic movement practice, ADL, and over-ground walking practice. In addition to traditional physical therapy, the patients underwent walking assessments and training using the Medical Care Pit: HPD-BT03 (Cyberdyne Inc., Tsukuba, Japan) twice a week, totaling 4 times over 2 weeks (Figure 1). The patients performed walking assessments with MCP only at the 1st and 4th sessions. Finally, the last period was a post-assessment period where patients performed post-assessment tasks as in the pre-assessment period (Figure 2).

### 2.4. Intervention (Medical Care Pit Walking Support System)

Walking assessments and training with MCP were performed in 20-min periods, including rest, with each day counted as one session (excluding set-up time). Patients were firmly secured to trunk and pelvic harnesses to prevent falls or partially reduce body weight effects (unloading). The physical therapist adjusted the amount of unloading according to the posture of the patients; amounts were set to the minimum possible level required. In particular, the physical therapist checked the gait posture, reduced paretic single-leg support time, and the back of the knee or collapse of the paretic limb in the gait cycle. The physical therapist provided visual feedback via a monitor to the patients about the symmetry of step length and toe clearance during walking. Videos could be recorded for review or demonstration purposes (Appendix A).

### 2.5. Outcome Measures

All patients underwent pre- and post-assessments by the same certified physical therapist. The primary outcome measure was the feasibility (specifically, the completion rate of walking training for 2 weeks using the MCP). The completion rate was defined as the number of persons who were able to implement the 2-week protocol within 2 weeks ± 5 days, divided by the total number of persons.

The secondary outcome measures were the Fugl–Meyer Assessment-Lower Extremities (FMA-LE) for motor paralysis [11], Functional Ambulation Category (FAC) for independent walking [12], gait parameter on the treadmill (steps, step length, step length symmetry, cadence, and treadmill speed) for walking ability, the training burden on therapists while utilizing MCP, satisfaction levels for both therapists and patients while training and assessing with MCP, Modified Borg scale, actual required time and manpower (number of persons) during preparation, wearing, and gait analysis with MCP, and adverse events for safety. The FMA-LE was chosen due to recommendations by the Japanese guidelines for the management of stroke, in addition to its resolution in capturing recovery stages compared to other, popular measurement protocols, such as the Brunnstrom recovery stages [13,14]. In terms of the questionnaires, both the physical therapist and the patient completed the following five questionnaires, selecting the answer to each question from “1. very good”, “2. good (rather satisfied)”, “3. neither good nor bad”, “4. bad (somewhat dissatisfied) “, and “5. very bad” in relation to Q1–3. Also selected were Q4–5 from 0 (no anxiety) to 10 (severe anxiety). The lower the score for these responses, the higher the level of satisfaction or the lower the level of anxiety.

(Q1)What is your satisfaction with this device throughout the whole? (Q2)What is your satisfaction with the gait assessment using this device?(Q3)What is your satisfaction with the gait training using this device?(Q4)What is your anxiety about the gait training using this device?(Q5)What is your anxiety about falling when you use this device?

### 2.6. Criteria for Discontinuation (Discontinuation of Protocol Treatment)

(1)The patient requested to withdraw from the study or withdrew consent.(2)An event that met the exclusion criteria occurred during the study period.(3)When a disease or other condition developed that made it difficult to continue the research.(4)When a patient’s medical doctor judged that discontinuation was appropriate, based on the evaluation of efficacy or assurance of safety.(5)The discontinuation of the entire study as a whole.

### 2.7. Ethical Statement

This protocol was approved by the University of Tsukuba Clinical Research Review Board (TCRB 18-005) and registered with the Japan Registry of Clinical Trials on 1 October 2018 (jRCTs032180020). The participants, either themselves or with appropriate substitution, provided their written, informed consent to participate in this study.

### 2.8. Statistical Analysis

For statistical analysis, in addition to descriptive statistics (mean, standard deviation, etc.) for participants, the Wilcoxon signed-rank test was used to assess outcomes before and after the intervention. The significance level was 5% on both sides and the confidence level was set to 95%. All statistical analyses were performed using IBM SPSS version 26.0 (IBM Corp., Armonk, NY, USA).

## 3. Results

### 3.1. Patient Characteristics

Since our hospital is an acute facility, many patients were transferred to rehabilitation hospitals (including nursing homes), within a few days after acute medical treatment. Therefore, we intended to focus on a feasibility study with a few patients using MCP performed in the Center for Innovative Medicine and Engineering (CIME) located in our university hospital.

As a result, a total of eight patients, five of whom were female, participated in this study. Of the eight total, three patients had ischemic stroke while five had hemorrhagic stroke and five were experiencing paresis on the right side while three had left-side paresis. Patients’ mean age was 60.7 ± 16.3 years and time since stroke occurrence was a mean of 14.1 ± 6.5 days. Patient characteristics are summarized in Table 1.

### 3.2. Completion Rate and Adverse Events

All patients completed this study over a total mean period of 10.5 ± 1.6 days. No adverse events were reported.

### 3.3. Improvements in Functional Parameters

Although motor paralysis scores did not significantly improve over the study period, the independent walking score increased from 2.2 ± 0.8 to 3.1 ± 1.3 (*p* = 0.020) as measured by the FAC instrument (Table 2). Both step lengths and treadmill speed showed significant improvements over the study period while patient burden, as measured by the modified Borg scale, significantly decreased (2.7 ± 1.6 to 2.0 ± 1.6; *p* = 0.014). Pre- and post-assessment parameters are summarized in Table 2. 

### 3.4. Qualitative Evaluation of MCP Use and Features

The actual time and manpower (number of persons) during preparation, wearing, and gait analysis with MCP were about 1–2 min and one person. The physical therapist was careful to avoid falls as patients ascended and descended the treadmill; one out of eight patients moved onto the treadmill in a wheelchair using a lightweight ramp for safety. The physical therapist and patient questionnaires for the MCP are summarized in Table 3. Satisfaction with gait assessment and training with the MCP was high for both the physical therapists and patients. In particular, the anxiety scores for the physical therapist regarding gait training and falling with MCP significantly decreased (3.8 ± 2.3 to 1.0 ± 1.6; *p* = 0.027 and 3.1 ± 2.2 to 0.8 ± 1.3; *p* = 0.045) from the first to the fourth sessions (Table 3). 

In terms of therapist and patient burdens and satisfaction while training and assessing using MCP, the following comments were made in the questionnaire survey about the hardware and software for the MCP:Handrails and harnesses (trunk belts) can be used to safely implement walking exercises in the acute period.Gait measurement (left and right stride length) can be carried out conveniently.Objective values can be easily calculated, and regular evaluations can be carried out in the acute period.Easy feedback to patients on the results of gait measurement.

In terms of problems/improvements, patients found operation buttons to be complex, the location of the emergency stop button had to be re-examined, and communication errors occurred between gait cameras and computers. Several patients commented that the use of handrails provided them with a sense of safety and security. Other comments included ‘it would be nice to have it in the house’, ‘it would be nice if music could be played during walking practice’, and ‘it would be nice to see nature (e.g., scenery or images)’.

## 4. Discussion

Here, we report an interventional analysis of the utility of rehabilitation with a walking support system for acute stroke victims. We found that MCP (an assistive rehabilitation device), in addition to the usual rehabilitation program, improved functional parameters for patients while reducing perceived burden over the study period. The customizable nature of the system, coupled with real-time reporting capabilities to provide step count, distance, etc. to both staff and patients, resulted in significantly improved outcomes for participants while somewhat improving burdens of caregiving staff (albeit not significantly) and significantly decreasing anxiety for the physical therapist regarding falls. Combined with traditional therapy, the MCP system should serve as a useful support device for acute stroke patients.

In general, patients with acute stroke perform physical, occupational, and/or speech-language-hearing therapies depending on their symptoms [15]. Many stroke patients often want to walk again; therefore, the main goal of physical therapy is usually to acquire independent walking [15]. For this reason, there are some focused interventions for developing walking power and ability in patients with stroke, such as body weight-supported treadmill training (BWSTT; introduced in the 1990s), that have reports of functional improvements [4,16]. However, a Cochran Review reported that ambulatory patients with stroke significantly improved their walking speed and walking endurance after such treadmill training, both with or without body weight support [17]. Subsequently, electromechanical and robot-assisted walking devices, mainly end-effector types (e.g., Gait Trainer) and exoskeleton types (e.g., Lokomat), have been used in stroke rehabilitation since the 2000s to reduce therapist manual assistance burdens [18,19]. It has also been reported that non- or low-ambulatory ability patients with early after-stroke therapy significantly improved their independent walking after mechanically assisted rehabilitation compared with normal care [20]. In particular, patients within a 3-month post-stroke window and those unable to walk may experience benefits from this intervention type [20]. These interventions are recommended by the AHA/ASA Guideline and the Japan Stroke Society Guideline 2021 for the Treatment of Stroke [6,13]. Additionally, from a research standpoint, the introduction of gait devices and robotic technology into stroke rehabilitation has the potential to simultaneously capture data from multiple functional parameters, increasing the resolution of progress evaluation.

Mechanically assisted therapy combined with usual rehabilitation may be useful since there are some issues with current traditional gait assessment and training or BWSTT as follows: It is not easy to provide real-time feedback to the physical therapist and patients using objective gait parameters while visual feedback of gait analysis (e.g., step-length symmetry and toe clearance during walking) is delayed by analysis time.In general, numerous medical staff and sufficient laboratory space are required to obtain gait parameters in walking-dependent patients with acute stroke. It is expected that equipment which measures the posture of a stroke with a camera in a simple way will be introduced into clinical practice and simplify this step [21,22].Physical therapists provide assistance to stroke patients who require it, but the physical therapist’s own fatigue may interrupt the walking exercise. Since developing appropriate motor learning requires intensive repetition in sufficient amounts [2,23], physical therapists with many patients may not be able to meet this demand. There is also a need to focus on reducing the physical burden for long-term care providers, including family members, as well as for patients themselves.

The combination of the MCP system and usual rehabilitation retains several advantages over the traditional therapy systems utilized in clinical rehabilitation. First, session efficiency is particularly important in the acute phase when there is less time to spend with each patient. Currently, physical and occupational therapists must deliver advanced activities to facilitate ongoing integrated evaluation, in addition to planning and modification of treatments. Reports estimate that 52–74% of physical therapists and 70–79% of occupational therapists are currently engaged in such activities with inpatients in stroke hospitals [24,25]. The introduction of technologies, such as robotics and artificial intelligence (AI), into clinical practice may improve the work efficiency of physical therapists and reduce both their physical and mental burdens. The MCP system’s tracking and recording capabilities can automatically and accurately record step length and walking distance, allowing gait training parameters to be precisely tracked over time. Second, feedback to the patient may contribute to setting specific goals and improve patient awareness and motivation. A previous study reported that feedback about gait performance once a day during routine physical therapy may optimize motor relearning after a stroke [26]. Thus, the MCP system’s sensors and recording abilities complement functional analyses as well as providing unbiased feedback to the patient.

Other MCP system advantages also exist. Stroke patients with severe hemiparalysis cannot perform hip and knee joint extension in the stance phase or hip and knee joint flexion in the swing phase. These can be in addition to knee collapse and backing of the knee due to gait posture in stroke patients [27]. Therefore, patients with stroke generally undergo gait training using knee-ankle-foot orthosis (KAFO) or ankle-foot orthosis (AFO) from the acute to sub-acute stages of stroke rehabilitation [28,29,30]. A previous study reported that the early wearing of KAFO was associated with higher recovery of ADL for stroke patients. However, when patients with severe stroke undergo gait training with the KAFO, this gait pattern is more likely to be performed with compensatory movements. Since from the acute phase of gait training, it is necessary to achieve a gait pattern that is close to normal, this could reduce progress towards independent walking or introduce problems in compensatory joints and muscles. For this reason, a previous study suggested that adding electrogoniometric feedback to standard physical therapy can improve the effectiveness of treatment for genu recurvatum (knee hyperextension) in stroke rehabilitation [31]. Recently, some researchers have developed gait training techniques that exploit a robotized KAFO to simultaneously assist ankle and knee joints [32]. For this reason, development of gait-assistive equipment and devices like MCP that approximate normal gait are needed for clinical applications. The utility of such equipment was shown in the current study, where the independent walking score (FAC), both step lengths, and treadmill speed showed significant improvements over the study period. This intervention (gait training with the MCP system), combined with innovative technology such as the wearable cyborg Hybrid Assistive Limb (HAL) system [33,34], may synergistically improve body function and walking ability in patients with acute stroke. Furthermore, there is also the possibility of stroke rehabilitation that combines a walking device with VR, augmented reality, and/or mixed reality in addition to upper limb training [35].

In terms of robot-assisted gait training (RAGT), it is crucial to combine RAGT with usual physical therapy in stroke patients [20]. One such device, the Gait Exercise Assistant Robot (GEAR) was developed to reduce the need for physical assistance and compensatory movements [36]. GEAR features precise assistance control and enriched feedback. Hirano et al. revealed in a case series that GEAR-assisted therapy improved the Functional Independence Measure score for walking (FIM-walk improvement efficiency) to 1.5 versus a historical control group (n = 15) score mean of 0.48 ± 3.2 (mean ± SD). GEAR is thus potentially useful for gait training in hemiplegic patients [36]. Technology such as GEAR contributes to rapid improvements in gait ability probably through knee flexion support on the afflicted side during the swing phase, increasing the training volume, plus a customizable stance/swing support mechanism for the afflicted limb, optimizing the training difficulty [37]. Although not directly comparable, patients with acute stroke could benefit from the MCP system’s partial unloading (via an independent unloading mechanism) with real-time gait feedback in this study. Such benefits of intensive, repetitive MCP practice with appropriate postural adjustment were evident in improved independent walking, step length, and treadmill speed while gait training without overload. Further comparable studies are needed to examine and to validate the effectiveness of the MCP system.

### 4.1. Limitations of MCP

While this device may be useful for stroke gait rehabilitation in that it easily calculates the stride, the number of steps, and the walking distance and feedback to the therapist and patient in real time, there is a need for further improvements in both hardware (e.g., operation buttons are complex, location of the emergency stop button needs to be re-examined) and software (e.g., communication errors between gait cameras and computers). Future revisions of the MCP software suite and hardware controls should improve familiarization times for both patients and staff. Recent advancements in electromyographic quantization and AI could also be integrated into future versions of MCP to provide full-body measurements of muscle activity for feedback-driven, real-time, ability-matched rehabilitation protocols that are individually tailored to each patient’s specific needs [8]. With sufficient miniaturization and scale manufacturing, MCP home units that combine daily life task assistance and exercises would be attractive for rehabilitation program monitoring and compliance.

### 4.2. Limitations of Study

We must acknowledge limitations. In this study, since MCP was used in concert with usual rehabilitation, we could not fully isolate and attribute functional improvements to only MCP effects in patients with acute stroke. For this purpose, further studies to fully delineate the specific benefits of MCP outside of normal rehabilitation support will be instructive. As a single-center study with a low number of enrolled patients, our results may not be generally applicable to broader populations. Additionally, long-term stroke survivors were not recruited and the ability of MCP to assist in functional rehabilitation after the optimal post-stroke recovery window was not assessed in this study. However, the flexibility of the MCP system would allow for easy integration into any ambulatory rehabilitation program. 

## 5. Conclusions

MCP is a useful device to support traditional rehabilitation therapy, improving functional ambulatory parameters even over a short (less than 2-week) period and reducing patient burden. All patients experienced improvements and no adverse events were reported. MCP is a safe, innovative platform amenable to the integration of newer advancements in rehabilitative technology and can be adjusted to the needs of any acute-stroke program. 

## 6. Patents

MCP is a patented device owned by the Cyberdyne Inc., Japan (Japan Patent Number 6054730).

## Figures and Tables

**Figure 1 jcm-12-05389-f001:**
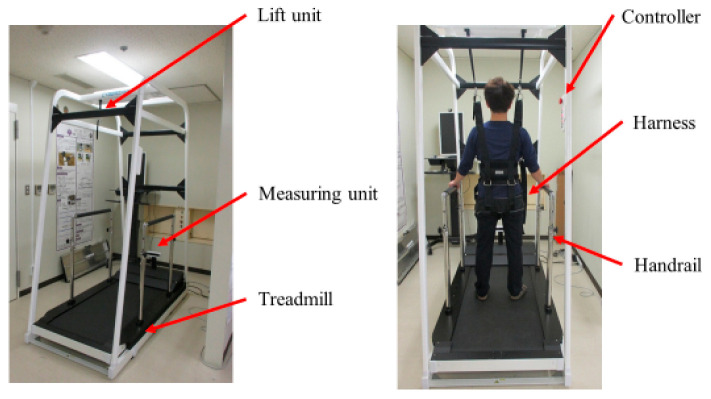
The Medical Care Pit walking support system.

**Figure 2 jcm-12-05389-f002:**
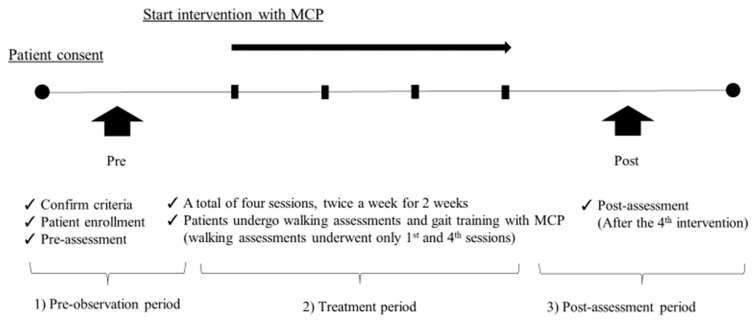
The study schedule. There were 3 periods in the study schedule. The first was the pre-observation period, the second phase was the treatment period, and the last period was the post-assessment period. MCP = Medical Care Pit.

**Table 1 jcm-12-05389-t001:** Demographic characteristics of patients who completed the study protocol.

Age (Years)		60.7 ± 16.3
Sex	Men	3
	Women	5
Height (cm)		160.1 ± 5.6
Weight (kg)		65.7 ± 14.3
Type of Stroke	Ischemic	3
	Hemorrhagic	5
Side of Paresis	Right	5
	Left	3
Time Since Stroke (d)		14.1 ± 6.5
Study Period (d)		10.5 ± 1.6

Values are mean ± SD.

**Table 2 jcm-12-05389-t002:** Pre-therapy vs. post-therapy assessment comparison.

Evaluation Index		Pre	Post	*p*
Motor Paralysis	FMA-LE	21.1 ± 7.2	22.6 ± 6.5	0.106
Independent Walking	FAC	2.2 ± 0.8	3.1 ± 1.3	0.020
Gait Measurement	Right Step Length (m)	0.24 ± 0.07	0.36 ± 0.07	0.025
	Left Step Length (m)	0.21 ± 0.10	0.35 ± 0.08	0.017
	Step Lengh Symmetry (Paretic/Non-Paretic)	1.70 ± 1.51	1.04 ± 0.11	0.263
	Cadence (Steps/Min)	86.6 ± 48.0	89.2 ± 40.3	0.674
	Treadmill Speed (km/h)	1.1 ± 0.1	1.6 ± 0.5	0.018
Gait Training Burden	Burden on Patient (Modified Borg Scale)	2.7 ± 1.6	2.0 ± 1.6	0.014
	Burden on Medical Staff (Modified Borg Scale)	1.5 ± 0.9	1.0 ± 0.7	0.167

Values are mean ± SD; FMA-LE: Fugl-Meyer Assessment-lower extremities, FAC: Functional Ambulation Category; Wilcoxon signed-rank test.

**Table 3 jcm-12-05389-t003:** Qualitative evaluation of the MCP using the five questionnaires.

Evaluation Index	1st Session with MCP	4th Session with MCP	*P*
From Physical Therapist			
Q1: Satisfaction with MCP (Whole)	2.6 ± 1.3	1.8 ± 0.8	0.216
Q2: Satisfaction with MCP (Gait Assessment)	2.2 ± 1.0	1.6 ± 0.5	0.096
Q3: Satisfaction with MCP (Gait Training)	2.1 ± 0.8	1.7 ± 0.4	0.180
Q4: Anxiety with MCP (Gait Training)	3.8 ± 2.3	1.0 ± 1.6	0.027
Q5: Anxiety with MCP (Falling)	3.1 ± 2.2	0.8 ± 1.3	0.045
From Patient			
Q1: Satisfaction with MCP (Whole)	1.8 ± 0.6	2.0 ± 0.7	0.317
Q2: Satisfaction with MCP (Gait Assessment)	1.8 ± 0.6	1.8 ± 0.6	1.000
Q3: Satisfaction with MCP (Gait Training)	1.7 ± 0.7	1.8 ± 0.6	0.317
Q4: Anxiety with MCP (Gait Training)	2.0 ± 2.4	0.7 ± 1.0	0.068
Q5: Anxiety with MCP (Falling)	1.3 ± 2.4	0.7 ± 1.0	0.257

Values are mean ± SD. MCP: Medical Care Pit. Wilcoxon signed-rank test.

## Data Availability

This study registered with the Japan Registry of Clinical Trials on 1 October 2018 (jRCTs032180020).

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
