# Peer review of "Safety and Feasibility Study of the Medical Care Pit Walking Support System for Rehabilitation of Acute Stroke Patients"

_jcm, 2023, doi:10.3390/jcm12165389_

Round 1

Reviewer 1 Report

This paper proposes that evaluate the stroke patients' gait and real-time independent walking score by using Medical Care Pit (MCP) to support stroke patients' rehabilitation. However, the following issues must be addressed.

1. The recent situation regarding stroke rehabilitation is mentioned in the introduction. However, the lack of hand exoskeleton rehabilitation is not tolerated. Hand rehabilitation is a giant research area. Thus, recent works such as “Development of an Untethered Adaptive Thumb Exoskeleton for Delicate Rehabilitation Assistance, in IEEE Transactions on Robotics, vol. 38, no. 6, pp. 3514-3529, Dec. 2022” should be cited for recent development.

2. In this paper, the section 2, part of section 3, and part of section 4, the fonts are italic. Please be advised, the fonts should be consistent.

3. The Fugl-Meyer is scored by therapists’ subjective view. The different therapist has different score for the same patient's gait situation. Therefore, please discuss why the evaluation method is selected and whether there is a better method. 

Minor editing of English language required.

Author Response

Response to Reviewer 1 Comments

Thank you very much for your careful review. We have listed our responses under each comment.

Reviewer 2 Report

Interesting study using new technologies in the rehabilitation of stroke patients.

The increase in demand for rehabilitation treatments makes it relevant to introduce new technologies that allow maintaining the quality of the intervention with a reduction in costs.

Simple study (including training with the "Medical Care Pit" in the treatment of patients). Due to the poor methodological design, it remains doubtful whether the study intends to evaluate the feasibility of this method or whether it intends to verify the functional gains of stroke patients (impossible to determine because interventions are used together with no control arm).  This point should be clarified to give robustness to the study.

There are a few issues that need clarification:

P1

Abstract:

- The use of the "Medical Care Pit" device was carried out in addition to the conventional physiotherapy treatment, so the evaluation of the results must be done together. This must be very clear. We do not know if the improvements presented by the patient have anything to do with the Medical Care Pit or the other treatments.

L-31/32 “Taken together, the MCP was effective in gait rehabilitation towards independent walking and relieved burdens on the patients.” It would be clearer: in addition to the usual rehabilitation program

P2

Materials and Methods

Participants

L-67/68 - “Eight patients with acute stroke were admitted to the University of Tsukuba Hospital between October 2018 and December 2019.” Only eight stroke patients were admitted during this period or eight were included in the study?

L-80 “7) those whose medical doctors decided that participation was not appropriate” this is not clear, should be specified.

P4

Results

3.1. Patient Characteristics

Assuming that only eight were included in the study (by application of the inclusion and exclusion criteria), it was important to know how many patients were considered for the study and why they were excluded.

P5

Improvements in Functional Parameters

L-155 – “Left step length” or step length?

Qualitative Evaluation of MCP Use and Features

L-160 – In the Study schedule a questionnaire survey was not indicated. It should have described what it evaluated in that questionnaire survey. It should also be broken down what were the comments of patients and physiotherapists.

P 6

Discussion

L-182 - “We found that MCP, an assistive rehabilitation device, improved functional parameters for patients while reducing perceived burden over the study period.” We don't know if the MCP improved functional parameters. The MCP and the rehabilitation program improved functional parameters.

L- 186 - “while somewhat improving burdens of caregiving staff”

It is not noticeable how it has been improving burdens of caregiving staff and how it was measured. It appeared that the patients were doing the usual rehabilitation program along with the MCP . Or were there treatments that were replaced by the MCP

P7

L-220 -  “Physical therapists provide assistance to stroke patients who require it but the physical therapist’s own fatigue may interrupt the walking exercise. Since developing appropriate motor learning requires intensive repetition in sufficient amounts [2,21], physical therapists with many patients may not be able to meet this demand.” It is only significant if training with MCP replaces the usual treatment (it was not done in this study), so there is no evidence that only MCP it has significant functional improvements.

L - 227-232 “The MCP system retains several advantages over the traditional therapy systems utilized in clinical rehabilitation. First, session efficiency is particularly important in the acute phase when there is less time to spend with each patient. Currently, physical and occupational therapists must deliver advanced activities to facilitate ongoing integrated evaluation, in addition to planning and modification of treatments. Reports estimate that 52-74% of physical therapists and 70-79% of occupational therapists are currently engaged in such activities with inpatients in stroke hospitals [22,23].” But there is no evidence that only MCP it has significant functional improvements.

P8

Limitations of Study

It should be noted that MCP it was not used alone but combined with the rest of the rehabilitation program. In this way, we do not know if the functional results obtained are related to the MCP.

Author Response

Response to Reviewer 2 Comments

Thank you very much for your careful review. We have stated our response under each of your comments.

Round 2

Reviewer 2 Report

Congratulations on the article. Satisfactory response to all questions raised